# Experimental Study on Fire Sources for Full-Scale Fire Testing of Simple Sprinkler Systems Installed in Multiplexes

Jeonghwa Park [1] and Jihyun Kwark [2,*]

1 Department of Safety Engineering Graduate School, Seoul National University of Science and Technology, Seoul 01811, Korea; jhp11982@seoultech.ac.kr
2 Fire Insurers Laboratories of Korea, Gyeonggi-do 12661, Korea
* Correspondence: kwark@kfpa.or.kr; Tel.: +82-10-6231-5343

**Abstract:** Fires are accidents that can cause numerous human casualties in multiplexes. The simple sprinkler systems applied in South Korea employ sprinklers to protect people against residential fires, as specified by the National Fire Protection Association (NFPA) standard 13D. Therefore, it is necessary to evaluate the fire control performance of multiplexes, which are at a greater risk than residential facilities. This study aims to verify the fire control performance of simple sprinklers in multiplexes and to develop a fire source that can be used as a protocol for testing fire suppression methods. The fire source was evaluated by using a 3 MW large-scale calorimeter (ISO 13784). The proposed fire source for multiplexes was applied in various forms according to the application methods, with ignition sources including cotton wick, wood crib, and heptane, and then the fire tests were conducted.

**Keywords:** full-scale fire test; multiplexes; fire source; simple sprinkler system

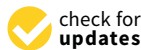



## 1. Introduction

According to the National Fire Information System of the National Fire Agency in South Korea, 44,103 (approximately 10%) of the total 426,521 fire cases in the last ten years (2010–2019) [1] occurred in areas with multiplexes. Many human casualties have been reported as a result of fires in different types of multiplexes such as karaoke bars and room salons. This is because they provide low evacuation safety in the event of a fire due to the presence of rooms with partitions and narrow passages. Therefore, the risks for each business category must be identified, and reliable fire extinguishing systems with an appropriate fire suppression performance must be installed [2]. The simple sprinkler systems installed in multiplexes in South Korea are smaller than those typically used in buildings; moreover, the "domestic sprinklers" developed to provide the level of sensitivity, discharge rate, and water spray suitable for residential spaces fires (as specified by the National Fire Protection Association (NFPA) standard 13D [3]) have been introduced with their original structures. The purpose of domestic sprinklers is not to extinguish a fire, but to control a fire in order to facilitate the evacuation of residents. This is achieved by starting a spray of water in the rooms (including the upper walls) early in the event of a fire and lowering the temperature of the air currents around the ceiling and the breathing line temperatures to help the residents evacuate smoothly. Thus, sprinklers are protective systems developed to facilitate rapid evacuation and are thus suitable for small compartments.

According to the NFPA 13 [4], in the United States, depending on the use of space, fire hazard levels are mainly categorized as a Light Hazard (LH), Ordinary Hazard (OH), and Extra Hazard (EH). Classification is performed in a comprehensive manner, considering the fire energy density (which is a risk factor depending on the space), the heat release rate (HRR) of combustible material, and the historical effects of verified sprinklers [5]. Considering these hazard classes, among various types of multiplexes, "goshiwons" (very

small rooms that students live in while studying for important tests or when their normal homes are far from school) and postpartum care centers are used for residential spaces purposes. As such, they fall under the LH category. However, karaoke bars and room salons can be classified into the OH category because they present relatively high fire risks due to their various furnishing and finishing materials. For multiplexes, the extent of the damage due to a fire is not proportional to the area of the premises but is dependent upon the characteristics of risk with regard to the business category. Compared with residential facilities, multiplexes have higher fire loads or higher fire growth rates of combustible material; hence, many researchers have questioned whether domestic sprinklers are suitable for these types of premises [6]. Most of the studies on simple sprinkler systems have focused on supplementing existing systems [7–9]. Other studies related to sprinkler systems have focused on the interaction between fire and sprinkler spray [10–15] or sprinkler spray dynamics and modeling [16]. In addition, most studies on full-scale fire tests have examined the effect of sprinklers solely with regard to residential facilities. Other studies have been conducted on real-scale fire tests or the simulation of karaoke bars among multiplexes, but the fire characteristics were not considered in the sprinkler activation [17–19].

The fire control performance of domestic sprinklers is determined by examining the discharge rate and by using full-scale fire tests in accordance with the International Organization for Standardization (ISO 6182-10) [20].

In the case of South Korean standards for simple sprinkler systems, these tests are not performed, and fire control performance is indirectly examined through tests on the discharge rate and water spray. Therefore, fire tests are required to examine the fire control performance of simple sprinkler systems installed in multiplexes, which belong to a special business category in South Korea. Accordingly, customized fuel packages for simulating fire tests with fire loads and risks higher than those in residential buildings must be created to facilitate the development of reasonable test methods. This study aims to develop new fuel packages to identify fire hazards for multiplexes utilizing polyurethane foam (PUF) as the fire source in ISO 6182-10 fire tests. The feasibility of the developed fire source is verified by performing various full-scale fire tests using selected fuel packages.

## 2. Development of the Fire Source for Full-Scale Fire Tests

### 2.1. Selection of the Ignition Source

In this study, we attempted to develop a fire source for the fire testing of simple sprinkler systems installed in multiplexes by selecting standard combustible materials that reflect the fire risks in these types of buildings.

In this study, a field survey conducted on 31 multiplexes determined that the combustible material in karaoke bars and room salons mainly consisted of sofas, tables, monitors, sound systems, and air conditioners. A previous study performed a furniture calorimeter test by selecting furniture as the representative combustible material in residential facilities [21], classifying it according to the fire growth rates ($\alpha$) presented in the NFPA 72 (the National Fire Alarm and Signaling Code 2007, Annex B); sofas and drawers corresponded to "Fast" fire growth rates; chairs and tables corresponded to "Medium"; while home appliances, including televisions, corresponded to "Slow" rates. Therefore, in this study, sofas (with the highest fire growth rates) were selected as the standard combustible material in multiplexes. A sofa model was designed by referring to that used in fire tests conducted by Bwalya et al. [22] and Su et al. [7].

The European codes or technical standards are determined for design fire size by calculating the Heat Release Rate Per Unit Area (HRRPUA) for each occupancy (Table 1), and these can be used as the reference data [23,24]. Table 1 shows that a maximum HRRPUA of 250 kW/m$^2$ is suggested for dwellings or hotel rooms. For sales facilities such as shops, a slightly high HRRPUA ranging from 250 to 550 kW/m$^2$ is suggested due to the distribution of various combustibles. Fire conditions in fire experiments applying sprinkers take into account the harshest conditions. Therefore, the HRRPUA of combustible materials used in fire experiments was determined to be above 550 kW/m$^2$.

**Table 1.** HRRPUA for different occupancies [23,24].

| Occupancy | Heat Release Rate Per Unit Area (kW/m²) |
|---|---|
| Shop | 550 [21] 250 [22] |
| Offices | 290 [21] 250 [22] |
| Hotel rooms | 250 [21,22] |
| Dwelling, Hospital (room) | 250 [22] |
| Transport (public space) | 250 [22] |
| Industrial | 90–620 [21] |
| Library, theater (cinema) | 500 [22] |
| Excluding storage | Depending upon fuel arrangement [21] |

*2.2. Fuel Package Selection*

A fuel package, which is the ignition source to be used in fire tests to verify the fire control performance of simple sprinkler systems in multiplexes, was designed in the form of a sofa that corresponds to the fire size calculated in Section 2.1. According to ISO 6182-10, the fuel package in residential spaces fires uses a combination of wood crib and simulated furniture (Figure 1a,b). A three-person sofa was selected as the augmented combustible material for fires in multiplexes (Figure 1c). The sofa model was fabricated using six sheets of Polyurethane Foam (PUF) measuring 600 mm (W) × 600 mm (L) × 100 mm (D) [1EA] by referring to previous studies (Figure 2). Wood crib and a heptane can were used to ignite the combustibles (Figure 3). The specifications of each combustible material for constructing the fuel packages are as follows:

① Polyurethane Foam (PUF)
- 600 mm (L) × 600 mm (W) × 100 mm (D) [1EA]

② Wood crib
- 305 mm (L) × 305 mm (W) ×152 mm (H)
- Assembly weight: 2.5~3.2 kg

③ Heptane pan (for igniting the wood crib)
- 305 mm (L) × 305 mm (W) × 104 mm (H) (0.5 L of water and 0.25 L of normal heptane)

④ Cotton wick for igniting the sofa
- 150 mm (length) × 6.4 mm (diameter)

⑤ Heptane can (for igniting the sofa)
- 77 mm (inner diameter) × 153 mm (H) (80 mm (H) of water and 20 mm (H) of heptane)

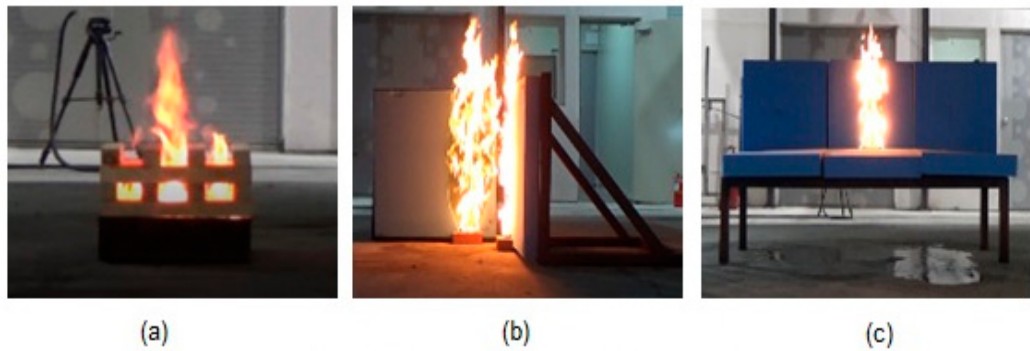

(a)　　　　　　　　　　(b)　　　　　　　　　　(c)

**Figure 1.** Independent measurement of heat release rate for (**a**) the wood crib, (**b**) polyurethane foam (PUF), and (**c**) the sofa model.

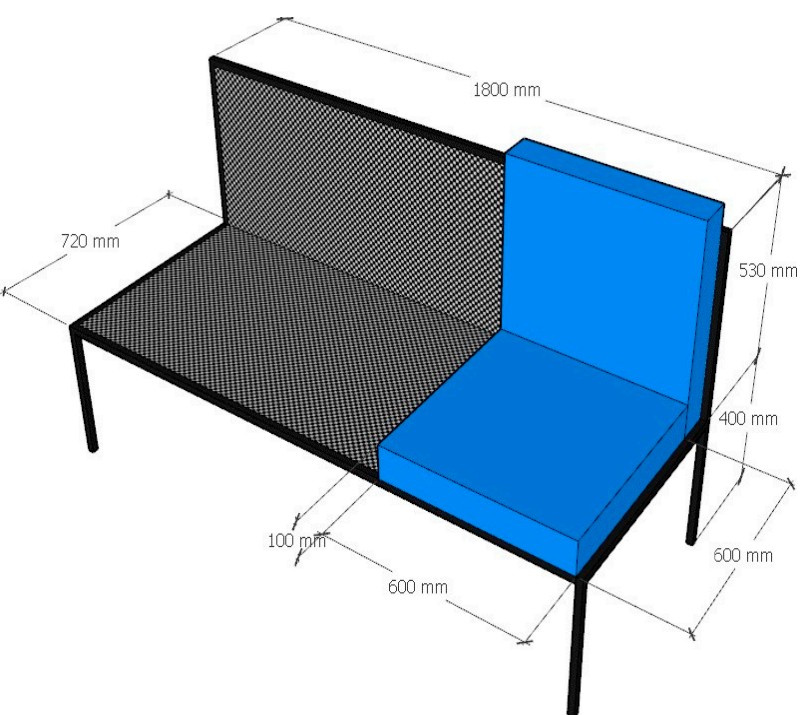

**Figure 2.** Sofa model used in fire testing.

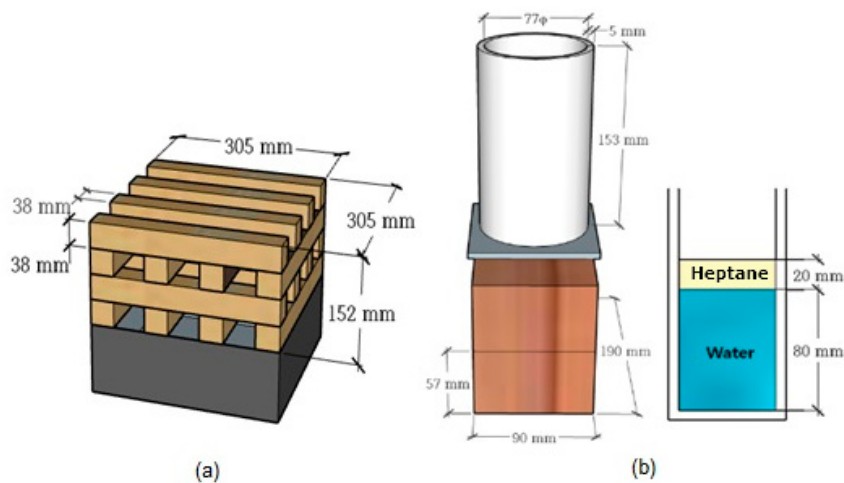

**Figure 3.** (**a**) Wood crib and (**b**) heptane can used in fire testing.

A cone calorimeter (ISO 5660) was used to measure the HRRPUA of the combustibles used for fire testing. We measured the HRRPUA of the PUF and wood cribs used in sofa models, as shown in Figure 4. The HRRPUA of the PUF used in the sofa model was 559.75 kW/m². This is relatively consistent with the HRRPUA of over 550 kW/m² as specified in Section 2.1.

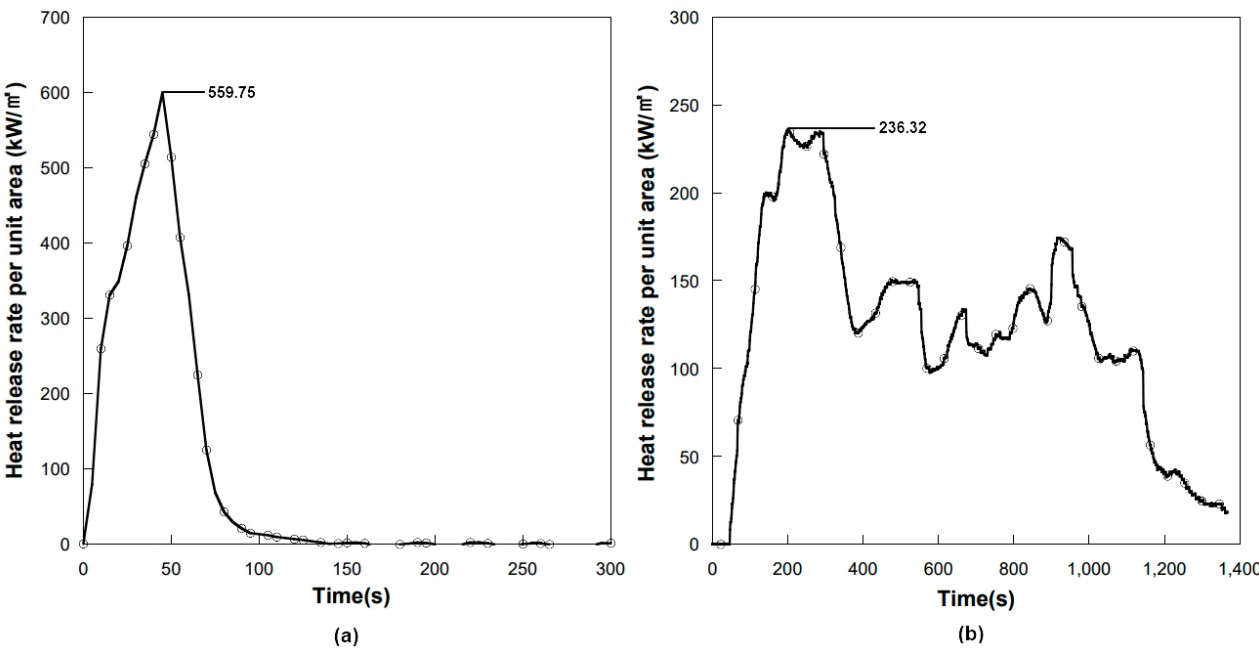

(a)

(b)

**Figure 4.** Heat Release Rate Per Unit Area (HRRPUA) for (**a**) polyurethane foam (PUF) and (**b**) Wood Crib.

The HRR of the wood crib and the simulated furniture were also measured to compare the fire sizes. A Large-scale Calorimeter 3 MW (ISO 13784) was used to measure the HRR (i.e., quantitative fire size) of the sofa model. Figure 5 shows the measured HRR of the three combustible materials. The peak heat release rates were 90 kW for the wood crib, 160 kW for the simulated furniture, and 360 kW for the sofa model. These results indicate that the HRR of the designed fuel package was higher than that of the fuel package for residential spaces.

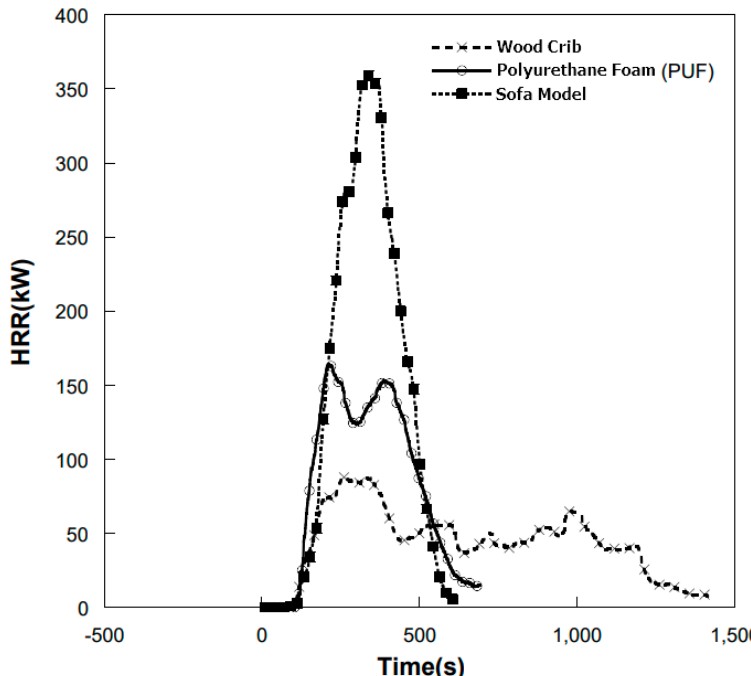

**Figure 5.** Comparison of the heat release rate (HRR) for the three fuels used in fire testing.

### 2.3. Preparation of a Fire Test Room and a Test Method for Full-Scale Fire Tests

2.3.1. Fire Test Room

The fire room of the standard ISO 6182-10 was used as the fire test room for the fire tests of simple sprinkler systems in multiplexes. The fuel package selected in Section 2.2 was used to reproduce the fire environment of multiplexes. According to the standard, the fire test room size was determined by the protection area of simple sprinkler systems. The standard number of simple sprinkler systems is two; hence, the fire test room size was determined to be the protection area occupied by two simple sprinkler systems (i.e., the protection width × double protection length). According to the National Fire Safety Code (NFSC), the radial distance of a simple sprinkler is 2.3 m. Thus, the actual protection width and length were 3 and 3.5 m, respectively. The fire test room size was 3 m (W) × 7 m (L). The ceiling height of 3 m, which was found to be the maximum height among the multiplexes surveyed, was applied to simulate a condition stricter than that of houses. Figure 6 shows the floor plan of the fire test room.

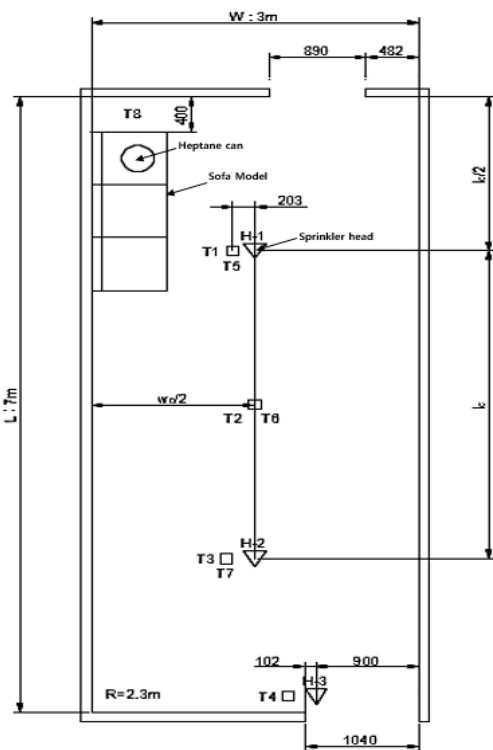

**Figure 6.** Schematic illustration of the fire test room and the thermocouple location. Note: T1–T4: Thermocouple 76 mm below the ceiling, T5–T7: Thermocouple 1600 mm above the floor, T8: Thermocouple 6 mm above the ceiling and 254 mm diagonally from the corner, H-1, 2, 3: Sprinkler head.

Meanwhile, non-combustible materials, such as gypsum boards, were used as the fire test room materials. For each test, dry plywood was attached to the two walls around the ignition source in multiple layers to facilitate fire spread through the walls and to simulate harsh conditions. Sheathed chromel–alumel K-type thermocouples with a 0.6-mm wire diameter were used for temperature measurements during the fire tests. A 10-channel analog-to-digital board (Graphtec GL240) was used as a temperature collection device. A commercial simple sprinkler certified by the Underwriters Laboratories (UL) and FM Global in the United States was used in the fire tests. Table 2 shows the simple sprinkler specifications.

**Table 2.** Specifications of the simple sprinklers used in fire testing.

| Item | Specification |
|---|---|
| Operating temperature | 72 °C |
| Maximum ambient temperature | 39 °C |
| Surface treatment | Nickel-chrome plating |
| Joint screw standard | 15 A (PT 1/2″) |
| Orifice size | Φ10.3 mm |
| Pressure for the pressure test | 2.5 MPa |
| Discharge rate | 50 LPM (0.1 MPa) |
| K factor | 50 ± 2.5 |
| Spray radius | R 2.3 m |

2.3.2. Fire Test Procedure and Test Method

The fire test procedure for verifying the fire control performance of the simple sprinkler systems in the multiplexes is as follows:

(a) First, the combustible material (i.e., ignition source) is ignited after fueling the heptane pan or can for ignition.
(b) The temperature is measured once or more per second using the temperature collection device.
(c) Firefighting water is supplied to the piping connected to the sprinklers, and the discharge pressure is set to 0.1 MPa.
(d) If the sprinklers begin to operate due to fire growth, the test is continued for 10 min.
(e) If the fire is not controlled by the discharged water, the test is immediately stopped, and the fire is extinguished using a fire hydrant.
(f) The water discharge is stopped after 10 min, and the remaining fire is extinguished manually.

2.3.3. Fire Control Performance Requirements

In fire tests, the performance of the fire extinguishing systems for fire control, such as sprinklers, is generally determined by the control of the heat air current temperature. In this study, the fire control requirements were determined as the heat air current temperatures around the ceiling and the breath line (1.6 m above the floor). In addition, the temperature of the ceiling material right above the ignition source was considered. The fire must be controlled with the two simple sprinklers; thus, the sprinkler installed near the entrance on the opposite side of the ignition source (H-3, Figure 6) must not operate. These requirements are summarized as follows, with the positions of each thermocouple for the temperature measurement shown in Figure 6:

(a) The maximum temperatures at points 76 mm below the ceiling must not exceed 315 °C.
(b) The maximum temperatures at points 1.6 m above the floor (height of the stream of breath) must not exceed 93 °C.
(c) The temperatures at the points described in item (b) must not exceed 54 °C for more than 2 min.
(d) The temperature of the ceiling material directly above the ignition source must not exceed 260 °C.

*2.4. Ignition Source Application Method for Constructing the Fire Source of the Fuel Package*

In constructing the fire source of the fuel package for fire testing simple sprinkler systems in the multiplexes, the fire growth characteristics can significantly depend on the ignition source application method. Therefore, the test conditions presented below were selected to be applied to the ignition sources and methods considering several fire situations. Figure 7 shows the fuel package under each of the following conditions.

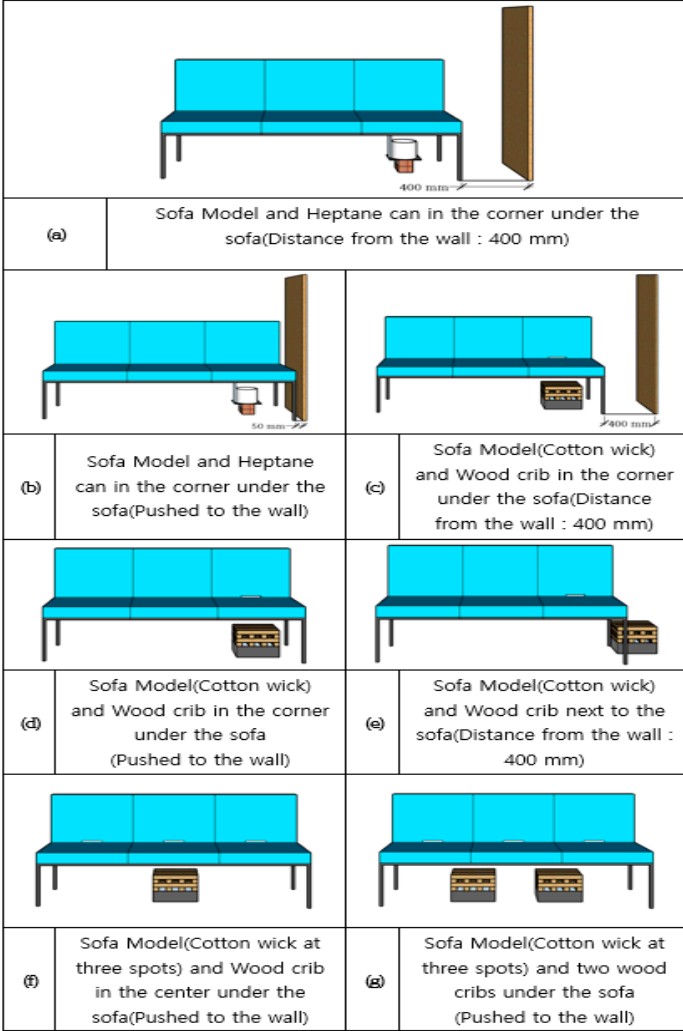

**Figure 7.** Application of various ignition sources to the construction of the fire source of the fuel package.

(a) Practically the most feasible fire scenario, in which a fire expands and spreads to the bottom of the sofa after a trash can or a cigarette butt under the sofa catches fire. The heptane can is used to simulate a small ignition source. The sofa is placed in the corner of the fire room and separated from the wall by the seat depth (400 mm) considering the condition that another sofa is placed next to it in the perpendicular direction.

(b) The same ignition method as (a) is applied, but the sofa is pushed up against the wall, to consider the case of using one sofa to simulate a condition favorable for fire growth.

(c) The wood crib is used as the ignition source, instead of a heptane can, in the corner under the sofa. A combustible material much larger than a trash can (e.g., stove) is simulated. The sofa is separated from the wall by 400 mm.

(d) The condition is the same as that in (c), but the sofa is pushed up against the wall to simulate a more severe condition.

(e) The sofa is placed 400 mm away from the wall. The wood crib, as the ignition source, is placed next to the sofa, not under the sofa, such that it can be reached by the firefighting water from the sprinkler (uncovered condition).

(f) The condition is the same as that in (c), but the wood crib is placed in the center under the sofa, and cotton wicks are placed on the sofa. The wood crib and the cotton wicks are ignited at the same time.

(g) The condition is the same as that in (f), but two wood cribs are used as the ignition sources.

## 3. Full-Scale Fire Test Results and Considerations

### 3.1. Sofa Model Fire Test Results and Discussion According to the Ignition Sources

Commercial simple sprinklers certified by UL and FM in accordance with ISO 6182-10 were used in the fire tests to examine the validity of the developed fuel package to verify the fire control performance of the simple sprinkler systems in multiplexes (Section 2.3.1). These products can properly control the heat air current temperature in the fire test of the combination of the wood crib and simulated furniture (i.e., existing fuel package). In this section, fire tests were conducted under various strictly selected conditions, as shown in Figure 7 (Section 2.4), to select a reasonable fuel package ignition method that meets probable fire scenarios.

Figure 8 shows a scene where the maximum temperature is reached in the fire test according to the ignition source application method shown in Figure 7.

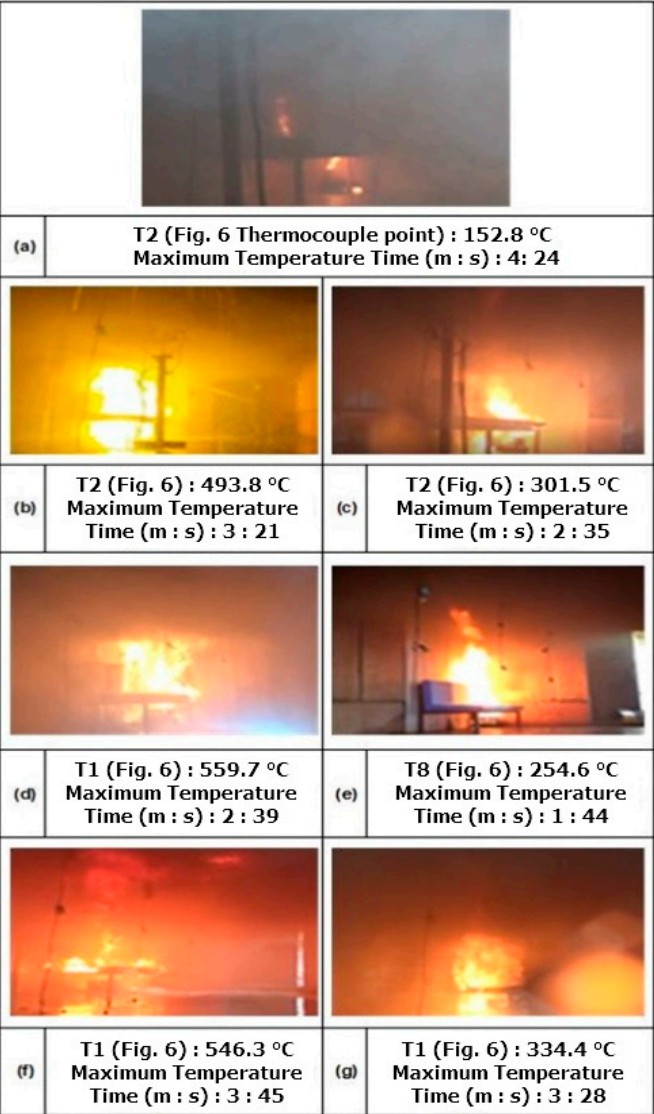

**Figure 8.** Scene when the maximum temperature is reached in the fire test according to the ignition source application plan.

The fire test results showed that the fire growth characteristics varied depending on the type of ignition source. When the wood crib, a powerful ignition source, was used, or when the sofa model was pushed to the wall in the tests that simulated a covered fire for which the ignition source of the sofa model was placed under the sofa, the fire spread to the entire room despite the operation of the sprinklers because fire suppression was delayed or fire control failed in the worst case. This is because the water from the sprinklers could not reach the covered area under the sofa, even though it delayed the fire growth by cooling the surface of the sofa model. As the fire caused by the ignition source continued under the sofa, it moved to the wall behind the sofa model and rapidly spread in the direction perpendicular to the ceiling. In other words, fire growth of the sofa model and the rapid spread of the fire to the wall are crucial variables for early fire control.

Among conditions (c) to (g), as shown in Figure 7, in which the wood crib was used as the ignition source, condition (c) failed because the third sprinkler installed near the entrance on the opposite side operated even though the temperature was controlled. All other conditions failed at fire control, except for condition (e), in which the wood crib was not covered, because the ignition sources covered under the sofa continuously burned even during the operation of the sprinklers, as mentioned earlier. The fire heated the wall behind the sofa, moved to the wall, and spread in the vertical direction to the ceiling. In particular, under conditions (f) and (g), in which the cotton wicks on the sofa were simultaneously ignited, the fire growth rate through the wall was further accelerated, making controlling the fire impossible.

Simple sprinkler systems are designed to perform proper fire control when they detect a fire and start operation before the fire grows to a certain size. They cannot suppress a covered fire because the penetration of water is difficult. Therefore, to secure the usefulness of simple sprinkler systems, additional efforts of the user are required to reduce fire risks as much as possible in terms of maintenance.

When the most likely fire scenario was applied (Figure 7a), fire control was possible with two sprinklers because the heat air current temperatures around the ceiling and the stream of breath were lower than the criteria in the three repeated tests, even though the fire spread to some parts of the wall along with sofa combustion. However, under condition (b), in which the sofa model was pushed to the wall, the flame rapidly spread along the corner between the two walls, making the heat air current temperature around the ceiling exceed the criterion at some points. Based on these results, condition (a) was selected as the optimal form of the fuel package and the ignition source for the multiplexes. In addition, the fire source of condition (a) was applied for further fire tests. Figure 9 shows the heat air current temperature around the ceiling over time according to the ignition method in the sofa model fire tests. In Figure 9, in conditions (d), (f), and (b) in which the sofa model was up against the wall and the ignition source was placed under the sofa model, the maximum temperature of the fire room reached 559.7 °C, 546.3 °C, and 493.8 °C, respectively.

However, condition (g) was not high compared to conditions (d) and (f) as the sprinkler was initially evolving while operating.

In the cases of (a), (c), and (e), in which the sofa model was not in close contact with the wall, the maximum temperatures of the fire room reached 152.8 °C, 301.5 °C, and 254.6 °C, respectively. In condition (c), the ignition source was located under the sofa model, and the maximum temperature in the fire room was relatively higher than that in condition (a) where the heptane can was used as the ignition source.

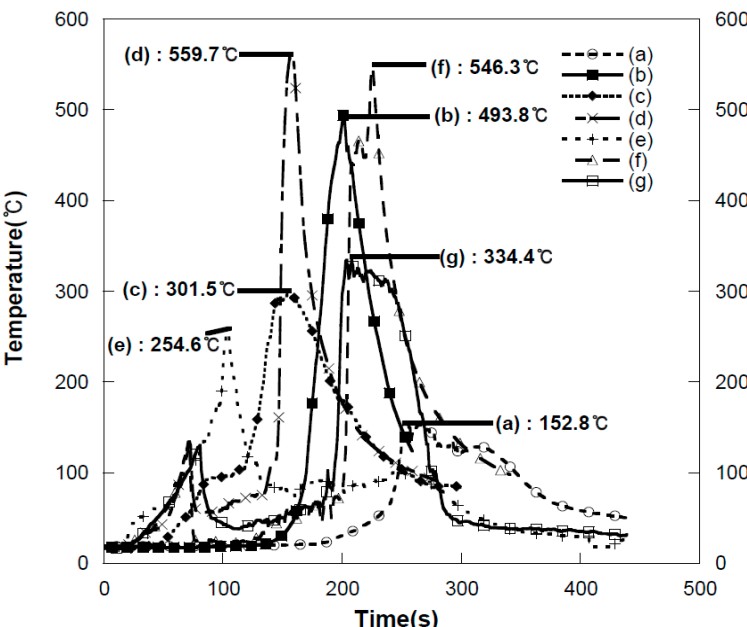

**Figure 9.** Ceiling heat air current temperature curves according to the ignition source in the sofa model fire tests.

### 3.2. Comparison of the Fire Control Performance with the Simulated Furniture Fire Test

In this section, the fire test results of the fuel package for houses were compared with those of the fuel package for the multiplexes for the commercial simple sprinkler systems used in Section 3.1 to examine the suitability of the newly developed fuel package. For each fuel package, the fire tests were repeated three times, and the heat air current temperature was measured, with the results shown in Figures 10 and 11.

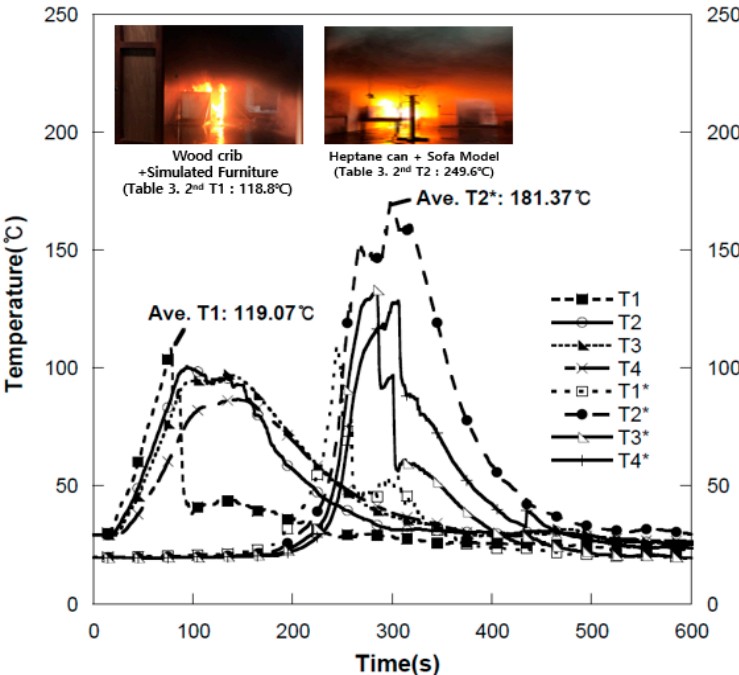

**Figure 10.** Comparison of the average heat air current temperatures around the ceiling in the fire tests of the fuel packages for residential spaces and multiplexes (*: fuel package for multiplexes).

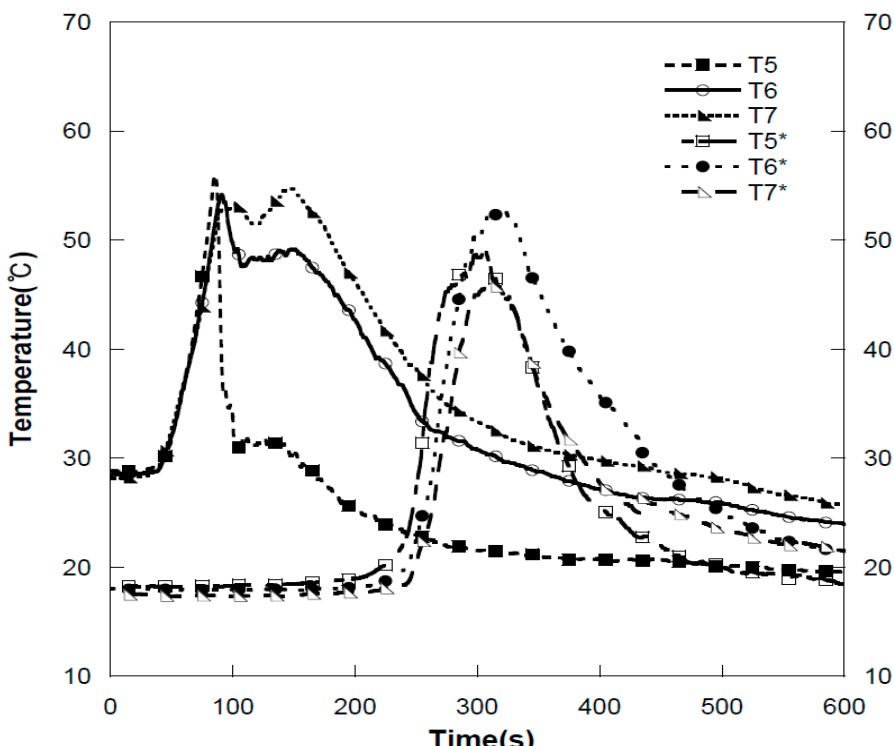

**Figure 11.** Comparison of the average heat air current temperatures around the stream of breath in the fire tests of the fuel packages for residential spaces and multiplexes (*: fuel package for multiplexes).

     First, the fuel package for the multiplexes that used the sofa model exhibited average ceiling heat air current temperatures that were measured using the three fire tests were up to 77 °C higher, compared with the fuel package for the residential space that used the simulated furniture at the same position (Figure 10). In the case of the sofa model fire, the heat air current temperature around the ceiling continuously increased even during sprinkler operation, indicating that the fire growth continued while the water was being discharged. However, in the case of the simulated furniture, the fire was easily controlled because the heat air current temperature was immediately reduced by sprinkler operation. This result clearly shows that the new fuel package has a higher fire growth rate. Meanwhile, the operation time of the sprinklers was different in the two fire tests because the simulated furniture exhibited fire growth immediately after its direct ignition with the cotton wick, but the sofa model showed fire growth after the fire was slowly spread by the heptane can under the sofa.

     The average heat air current temperatures around the breath line were similar (Figure 11). Although the heat air current temperature around the ceiling was slightly higher due to the relatively large growth of the sofa model fire, no significant difference was found in the heat air current temperature around the breath line due to effective fire control by the sprinklers. Table 3 shows the maximum temperatures of the two fuel packages at each position in the fire tests.

**Table 3.** Maximum temperatures by thermocouple position in the fire tests of the fuel packages for residential spaces and multiplexes.

| Fuel Package | | Maximum Temperature (°C) | | | | | | |
|---|---|---|---|---|---|---|---|---|
| | | **T1** | **T2** | **T3** | **T4** | **T5** | **T6** | **T7** |
| W.C. + S.F. | 1st | 118.4 | 102.9 | 94 | 82.8 | 59.8 | 46.2 | 59 |
| | 2nd | 118.8 | 108.1 | 115.4 | 99.5 | 64 | 66 | 68.1 |
| | 3rd | 120 | 102.1 | 99.2 | 87.9 | 44.3 | 54.6 | 45 |
| | Ave. | 119.07 | 104.37 | 102.87 | 90.07 | 56.03 | 55.60 | 57.37 |
| | | **T1 \*** | **T2 \*** | **T3 \*** | **T4 \*** | **T5 \*** | **T6 \*** | **T7 \*** |
| H.C. + S.M. | 1st | 132 | 152.8 | 139.2 | 130.1 | 51.1 | 53.3 | 46.4 |
| | 2nd | 133.6 | 249.6 | 151.9 | 146.7 | 52.2 | 54.3 | 48.7 |
| | 3rd | 134.1 | 141.7 | 126.3 | 116.5 | 48.2 | 53.1 | 46 |
| | Ave. | 133.37 | 181.37 | 139.13 | 131.10 | 50.50 | 53.57 | 47.03 |

\* Fuel package for multiplexes, W.C. + S.F. (Wood Crib + Simulated Furniture), H.C. + S.M. (Heptane Can + Sofa Model, for 3 persons).

### 3.3. Comparison of Fire Control Performances According to the Discharge Rate

To assess the validity of the fire source for the fuel package developed to verify the fire control performance of simple sprinkler systems in multiplexes, we attempted to observe whether fire control performance was maintained when water was discharged at pressures less than 1 bar (0.1 MPa), which is the minimum operating pressure of simple sprinkler systems. Fire tests were conducted using water discharged at 0.7 bars (0.07 MPa) which is 30% lower than the minimum operating pressure. The results were then compared with those of fire tests conducted at the specified pressure.

The discharge rate of a sprinkler was obtained as the product of the flow coefficient (K) and the square root of the discharge pressure ($\sqrt{P}$). K50 simple sprinklers are used in Korea; hence, approximately 42 L of water was discharged per minute at a pressure of 0.7 bars. This amount is approximately 8 L lower per minute than at the specified pressure, which significantly affects the fire control performance by reducing the cooling and fire extinguishing effect, which is the principle of sprinkler fire extinguishing (i.e., the fire suppression effect through the cooling action caused by the penetration of water was reduced by half).

In the three fire tests conducted at a pressure of 0.7 bars, the commercial simple sprinklers failed in terms of fire control, even though their performance was certified. Their performance could not overcome the fire growth rate of the sofa model, which was a new fuel package, and could not effectively block the spread of fire to the walls. The fire rapidly reached the ceiling via the corner of the fire room and the maximum heat air current temperature around the ceiling exceeded 600 °C (Table 4). The maximum heat air current temperature around the breath line also exceeded 100 °C, indicating that safe evacuation of the occupants was impossible. Figures 12 and 13 show a comparison of the average heat air current temperatures around the ceiling and the breath line, obtained in the three fire tests conducted at discharge pressures of 1 and 0.7 bars.

**Table 4.** Maximum temperatures by thermocouple position in the fire tests at a discharge pressure of 0.7 bars (0.07 MPa).

| Fuel Package | | Maximum Temperature (°C) | | | | | | |
|---|---|---|---|---|---|---|---|---|
| | | **T1 \*\*** | **T2 \*\*** | **T3 \*\*** | **T4 \*\*** | **T5 \*\*** | **T6 \*\*** | **T7 \*\*** |
| S.M. | 1st | 636.2 | 625.9 | 452.6 | 261.5 | 100.9 | 99.3 | 97.3 |
| | 2nd | 484.1 | 629.6 | 168 | 279.7 | 59.6 | 106.3 | 100 |
| | 3rd | 559.7 | 463.3 | 313.7 | 293.1 | 71.1 | 73.9 | 81.5 |
| | Ave. | 560 | 572.93 | 311.43 | 278.1 | 77.2 | 93.17 | 92.93 |

\*\*: 0.7 bar (0.07 MPa), S.M. (Sofa Model, for 3 persons).

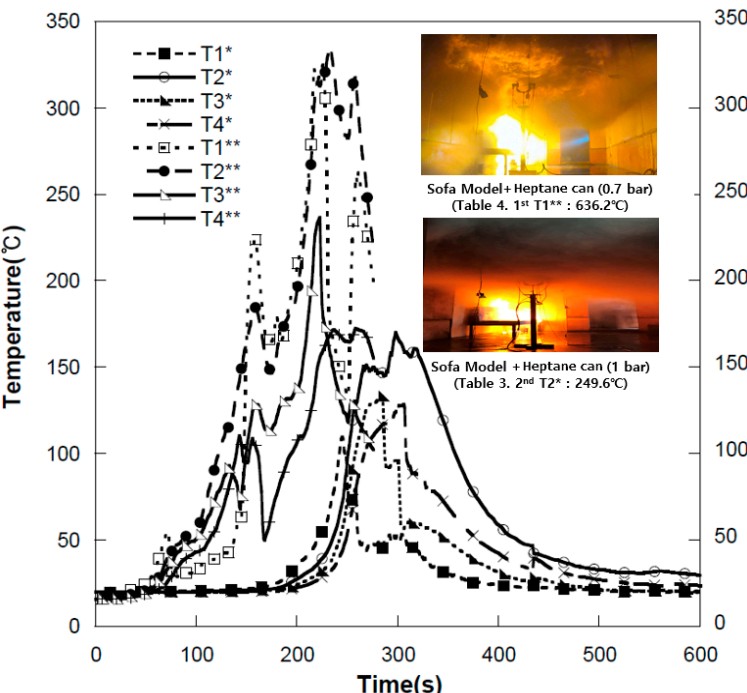

**Figure 12.** Comparison of the average heat air current temperatures around the ceiling in fire tests of the fuel package for multiplexes according to discharge pressure (*: 1 bar (0.1 MPa); **: 0.7 bars (0.07 MPa)).

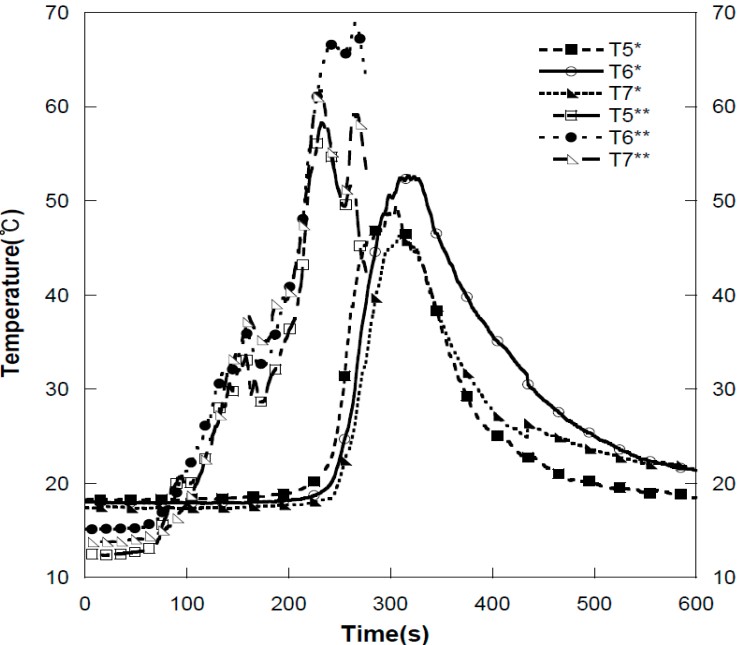

**Figure 13.** Comparison of the average heat air current temperatures around the stream of breath in fire tests of the fuel package for multiplexes, according to discharge pressure (*: 1 bar (0.1 MPa); **: 0.7 bar (0.07 MPa)).

The abovementioned fire test results showed that the form and size of the fuel package developed for the verification of the fire control performance of the simple sprinkler systems in the multiplexes were appropriately designed and valid. The proposed fire source could be used as a standard test method for performance certification or type approval of simple sprinklers installed in multiplexes.

## 4. Conclusions

In this study, the effectiveness of fire control performance of simple sprinkler systems installed in South Korean multiplexes has been studied by performing several experiments using a sofa model for three persons. The sofa model used a Large-scale Calorimeter 3 MW (ISO 13784), as per domestic and international fire test standards and prior research, and a test model with a Heat Release Rate (HRR) of 360 kW was developed.

The proposed sofa model was then used in various manners by applying seven different probable fire scenarios according to the type and arrangement of the ignition source. As a result, a scenario that successfully suppressed the fire using the simple sprinkler system at the minimum operating pressure (0.1 MPa) was selected. To reflect the harsh conditions of fire test, conditions that simulated a covered fire were applied, and the ignition source (a Heptane can) was placed under the fire source accordingly.

The fire control performance according to the change in discharge volume was compared to evaluate the fire control capability when a specific firefighting water pressure was not reached. Further, the suitability of the simple sprinkler head performance test as a fire source was reviewed.

Accordingly, when using the new fuel package with an increased fire size, the ceiling air flow was found to have a slightly higher temperature distribution than when using the fuel package for houses. This implies that the simple sprinkler applied to multiplexes should have a better firefighting performance than the existing residential sprinkler systems to obtain a sufficient level of fire control performance. In the future, we expect the fire source developed in this study to be used as a standard test method for performance certification or type approval of simple sprinklers installed in multiplexes, such as karaoke bars and general bars, which are at a greater risk of fire than residential spaces.

**Author Contributions:** Conceptualization, J.P., J.K.; methodology, J.P., J.K.; writing—original draft preparation, J.P.; resources, J.K.; writing—review and editing, J.P., J.K.; investigation, J.P., J.K.; supervision, J.K. All authors have read and agreed to the published version of the manuscript.

**Funding:** This study was supported by a research and development program (2018-NFA002-007-01010000) to improve the firefighting capabilities of the National Fire Agency.

**Data Availability Statement:** Data not available due to legal restrictions. (Due to the nature of this research, participants of this study did not agree for their data to be shared publicly, so supporting data are not available).

**Conflicts of Interest:** The authors declare no conflict of interest.

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
