# Peer review of "Experimental Study on Fire Sources for Full-Scale Fire Testing of Simple Sprinkler Systems Installed in Multiplexes"

_fire, doi:10.3390/fire4010008_

Round 1
Reviewer 1 Report
The authors propose a new calculation model as the one that is to be reliable when used to verify fire safety in rooms in a specific use. They calibrate its parameters in appropriately selected and quite carefully conducted experimental studies. It seems particularly valuable in these studies that many alternative fire development scenarios were considered and the results obtained were compared in a more or less rational manner. Both the presentations and discussion of the obtained results could be more careful, nevertheless, the proposed conclusions seem to be unambiguous and do not raise any doubts of the reviewer. The drawings could be drawn more precisely and a more detailed commentary could be added to them. Uppercase and lowercase letters are applied incorrectly in many places in the text. The recommended model is of great practical importance and its introduction to the relevant standards and regulations is undoubtedly advisable and expected, mainly due to the inadequacy of the computational approaches used so far.
Author Response
Dear Reviewer
Thank you so much for reviewing our manuscript.
We tried our best to respond to the reviewer's opinion.
Please see the attachment.

Reviewer 2 Report
The work that has been done is good and worth to be published. However, I believe that the manuscript should be focused on the experiments itself rather than pretending to extend the outcome of the experiments to a wider area, i.e. when the authors state “In this study, the effectiveness of fire control performance of simple sprinkler systems installed in South Korean multiplex available premises was verified by developing a suitable fire model and performing a fire test” is wrong to me and should be rephrased as “In this study, the effectiveness of fire control performance of simple sprinkler systems installed in South Korean multiplex available has been studied by performing several experiments using a sofa xxx…”
- The considerations presented about the HRRPUA (kW/m2) are not correct and need to be re-formulated. In particular when the authors mention the kW/m2 based on NFPA particular caution needs to be taken on the right meaning of those numbers that represent an average HRRPUA and not a HRRPUA for a given localized fire.
- It is not clear, for example, why the sofa chosen for the experiments is representative of a multiplex premise, wouldn’t be an upholstered sofa more hazardous?
- Heptane not heptan as often written in the manuscript.
- Line 94-104: there are arbitrary statements that might be right but that they are not supported by any evidence in the manuscript. Please extend this comment to all related statement in the manuscript.
- The use of words such as Fire models and Simulations easily induce the reader to expect a numerical analysis and this is not the case.
- References must be improved.
Author Response

(The authors gave the same response as above.)

Reviewer 3 Report
The manuscript presents a study on sprinklers in multiplex premises using experimental data. The reviewer recommends this work to be considered for publication after the following comments/corrections have been made:
- Please correct the term “Large-scale cone calorimeter” being used throughout the manuscript. Normally a cone calorimeter is referred to ISO 5660 while the data of the ignition source in the manuscript was more similar to a Large-Scale Oxygen Consumption Calorimetry (or Furniture Calorimeter). However very limited information about the instrument was available in the manuscript. They should be included in section 2.2.
- Literature review should be revised to consider recent development in multi-use buildings.
- Page 5, line 175: I believe “Factory Mutual “FM” should be “FM Global”
- Page 10, line 308: Discussion on difference between fire scenarios is missing
- Section 3.2: data on heat flux should be included for the comparison
- English regarding grammar and errors are recommended to be improved.
Author Response

(The authors gave the same response as above.)

Round 2
Reviewer 2 Report
Author's performed a comprehensive review of the mansucript according to the reviewers indications.